# Prevalence and Correlates of Anabolic–Androgenic Steroid Use in Australian Adolescents

**DOI:** 10.3390/nu17060980

**Published:** 2025-03-11

**Authors:** Dominic Byatt, Kay Bussey, Tim Croft, Nora Trompeter, Deborah Mitchison

**Affiliations:** 1Body Image and Eating Network, Discipline of Clinical Psychology, Graduate School of Health, Faculty of Health, University of Technology Sydney, P.O. Box 123, Broadway, NSW 2007, Australia; deborah.mitchison@uts.edu.au; 2Lifespan Health and Wellbeing Research Centre, School of Psychological Sciences, Macquarie University, Sydney, 2109, Australia; 3Graduate School of Health, University of Technology Sydney, Sydney, 2007, Australia; 4Population, Policy and Practice Research and Teaching Department, Institute of Child Health, University College London, London, WC1N 1EH, UK

**Keywords:** adolescents, anabolic–androgenic steroids, body image, muscularity, purging

## Abstract

Background: Within recent years, there has been a notable lack of research examining the factors associated with adolescent use of anabolic–androgenic steroids (AASs) in Australia, meaning information regarding risk factors of Australian adolescent AAS use is outdated and potentially inaccurate. Methods: To address this omission, the present study examined the prevalence and correlates of adolescent (aged 11 to 19 years) AAS use within the EveryBODY study, a large-scale representative survey of adolescents’ disordered eating behaviours and body image concerns, involving 5071 adolescents across thirteen schools within the Sydney and Newcastle/Hunter region of New South Wales, Australia. Results: A total of 1.1% of adolescents reported lifetime use of AAS to increase muscularity. In univariate analyses, increased prevalence of AAS use was associated with male sex (*OR* = 5.67), identifying as Aboriginal or Torres Strait Islander (*OR* = 3.80), identifying as same-sex or questioning sexual attraction (*OR* = 3.17), higher drive for muscularity (*OR* = 2.19) and weight/shape concerns in the past month (*OR* = 1.28), and higher frequency of purging (*OR* = 1.11) and binge eating (*OR* = 1.09) in the past month. In multivariate analysis, only drive for muscularity (*OR* = 2.44) and purging behaviours (*OR* = 1.10) remained as significant correlates. Finally, adolescents who reported lifetime AAS use also reported feeling significantly higher levels of distress and physical and psychosocial impairment compared to adolescents who reported never having used AAS to increase muscularity. Conclusions: Positive correlations between disordered eating and weight and shape concerns with AAS use suggests that adolescent AAS use may be conceptualised within the spectra of disordered eating among youth. These findings provide clinicians, carers, and educators with prototypical factors that should assist in the screening of adolescent AAS use to facilitate early intervention.

## 1. Introduction

Adolescence is a time of emotional turbulence, when teenagers are particularly vulnerable to sociocultural influences. Across genders, society has increasingly depicted the ideal body to be toned in musculature with very low amounts of body fat [1,2,3]. Increased exposure to these unrealistic body ideals has been linked to body dissatisfaction, including dissatisfaction with muscularity, and decreased self-esteem [4,5]. Body dissatisfaction has been demonstrated to peak in adolescence, where there is increased exposure to body ideals, scrutiny and self-consciousness regarding physical appearance [6]. Correlated with higher rates of body and muscularity dissatisfaction are behaviours adopted to “fix” perceived flaws in appearance that target the shedding of body fat and building of muscularity, such as dietary control, purging, use of performance enhancing supplements and weightlifting [7]. At the extreme end of these behaviours is the use of anabolic–androgenic steroids (AASs) [3,8,9], which is the focus of this study.

While effective at building muscularity, AASs are also a significant public health concern. AASs encourage muscle growth through protein synthesis, leading to significant increases in muscularity and strength when used with training [10,11,12]. On the other hand, AASs are associated with significant cardiovascular, psychiatric, and neurological risks, as well as increased risk of premature death [13]. Side effects have been reported in up to 94% of AAS users [14] and range from mood disorder symptoms and irritability [15,16,17] to increased risk of tendon rupture and left ventricular hypertrophy [14]. AAS dependence is also a common occurrence, with dependence rates as high as 23–32% being reported in some samples of AAS users [18,19]. Despite these known effects, most users believe AASs can be used safely long-term [14] and avoid reporting side effects to health practitioners due to stigma and perceived lack of knowledge [20,21]. This further increases the risk of harm as adverse consequences can go unmonitored. Due to these consequences, use of AASs for the purposes of increasing muscularity is now considered a significant public health concern [22,23], especially for vulnerable groups such as adolescents undergoing physiological development through puberty.

Although much of the current AAS literature has focused on adult males, the age of first AAS use is earlier than originally thought and includes a significant population of female users [24]. One meta-analysis, for instance, has found an overall lifetime prevalence rate of 2.3% for AAS use across international survey studies involving high school students [25], which aligns with Australian-based estimates (the setting for the present study) of 2.3% to 2.4% in 12–17-year-olds [26,27]. Although lower than the average rates of AAS use among adolescent males (6.4%), roughly 1.6% of females have reported using AAS within their lifetime [25]. Given that the side effects and dependence outlined above become more common and severe with age [14], it is imperative that efforts are made to intervene in AAS use early.

To facilitate health promotion, prevention, and early intervention efforts to reduce AAS use in young people, it is important to examine correlates of adolescent AAS use, as this assists in identifying vulnerable populations to target for efficient use of public health resources. In terms of demographic correlates, studies have demonstrated that male sex [7,26], older adolescent age [7,24], minority ethnicity including Indigenous young people [24,28,29], higher body mass index [7,30], and same-sex attraction [31,32] are associated with higher levels of AAS use among adolescents. Experts have postulated that these correlates may be due to mechanisms including a greater emphasis of muscular ideals within these demographic sub-groups and poorer general mental health related to lower mental health literacy, discrimination, and minority stress [33]. This is not to say that adolescents outside of these demographic profiles are immune. For instance, there has been an increasing focus on muscularity among female adolescents [1,34,35], with one study demonstrating similar prevalence rates of probable muscle dysmorphia (a condition characterised by intense dissatisfaction with one’s level of muscularity and linked to AAS use) across males and females in a 2017 survey of adolescents in Australia [36].

In terms of psychological correlates, greater body image dissatisfaction [15,24,29], higher drive for muscularity [24,37], and disordered eating behaviours such as purging and binge eating [29,32,38] have all been found to be associated with increased levels of adolescent AAS use. The clustering of AAS with such cognitive and behavioural features lends credence to the notion that AAS use could be considered within the spectra of disordered eating [39] and highlights that adolescents at risk for conditions such as eating disorders and muscle dysmorphia might also be at risk for AAS use. However, to assess if AAS use can be considered as a clinically significant psychiatric phenomenon on its own, the association between AAS use and both psychological distress and functional impairment also needs to be measured [40].

The present study aimed to provide updated information on the prevalence and correlates of adolescent AAS use in Australia, as the last Australian study examining factors associated with anabolic steroid use was based on data collection in 2005 [26]. Given the exploratory nature of this study, we did not have firm hypotheses in place for prevalence estimates, though we expected to replicate the correlates observed in previous studies. Secondly, given the deleterious effects of AAS and the association with disordered eating psychopathology previously documented, we did expect that AAS use would be associated with greater levels of psychological distress and functional impairment.

## 2. Methods

### 2.1. Sampling Procedure and Participants

Data were used from the first wave of the EveryBODY study, a longitudinal study of Australian adolescents’ disordered eating behaviours and body image concerns. The process of recruitment and data collection has already been described [41]. 

Briefly, between May and November 2017, 57 schools were contacted to participate in this study across the Newcastle/Hunter region and Sydney region in New South Wales, Australia, of which 20 schools agreed to participate. Reasons provided from the 37 schools who declined participation in this study included lack of time, limited staff availability, conflicting commitments, and participation in other research studies. Throughout the process of recruitment, seven schools withdrew their participation, citing conflicting commitments (*n* = 6) or concerns regarding questionnaire topics of dating and gender identity (*n* = 1). As a result, in 2017, surveys were administered to nine government schools and four independent schools (22.8% of all contacted schools). School enrolments ranged from 514 to 1305. The Index of Community Socio-Educational Advantage (ICSEA), which is a standardised measure of the educational advantage of schools based on their physical location, socio-economic background, and the occupation and education of students’ families, ranged from 909 to 1129 (*M* = 1035, *SD* = 60.91). As such, all ICSEA scores were within one standard deviation of the standardised mean across Australian schools (*M* = 1000, *SD* = 100).

All secondary students were invited to complete the survey, and a passive parental consent procedure was utilised (parents or guardians were given advanced and repeated notice of the study and could opt to have their adolescent not participate). Informed assent was also obtained from all participants. Students were given the online survey to fill out in class, which consisted of a range of questions including socio-demographic information, eating disorder pathology, weight and shape concerns, psychological distress, and quality of life impairment. This survey was pilot tested to ensure comprehension and completion within 50 min (i.e., a typical class period). Ethics approval was obtained from the University Ethics Committee (Macquarie University), the Catholic Education Office, and the New South Wales Department of Education. Information on response rates and sample size are included in the Results section below.

### 2.2. Measures

#### 2.2.1. Sociodemographic Questions

Participants were asked about their sex, age, height, and weight, which was used to calculate an age- and sex-adjusted BMI percentile score. One school within the sample refused to include separate questions on both sex and gender, and as such, their responses for ‘what is your gender: male, female’ were included within data for sex. Participants were asked whether they identified as Aboriginal, Torres Strait Islander, both, or did not wish to answer. Sexual attraction was assessed by asking participants whether they were attracted to boys (yes, no, or unsure) or girls (yes, no, or unsure), and participants’ sex was matched to their responses to classify them as ‘opposite-sex attracted’ or ‘same-sex attracted or questioning’. Finally, to examine first- or second-generation migrant status, participants were asked whether their parents were born overseas and were coded as either ‘no parents born overseas’ or ‘at least one parent born overseas’.

#### 2.2.2. Anabolic–Androgenic Steroid (AAS) Use

Lifetime AAS use was measured with the following question which was provided to participants, ‘Have you EVER used anabolic steroids to increase your muscularity? Remember your answers are confidential. Examples of anabolic steroids include testosterone enanthate injections and dianabol tablets’. If participants answered affirmative to this, they were classified as AAS users. Participants were also asked ‘Were these prescribed by a doctor?’.

#### 2.2.3. Weight and Shape Concerns

The averaged score of the items that comprise the Weight Concern and Shape Concern subscales of the Eating Disorder Examination Questionnaire (EDE-Q) [42] was used to measure weight and shape concerns reported by the participants. These items examine the severity of thinness-oriented weight and shape concerns over the past 28 days through 12 questions on a 7-point Likert-type scale, from 0 (no days/not at all) to 6 (every day/markedly), and includes questions such as ‘Has your shape influenced how you think about yourself as a person?’. Scores for the items across both subscales were averaged together and ranged between 0 and 6, with higher scores indicating more weight and shape concerns. The combined Weight and Shape Concern subscales have demonstrated excellent internal reliability for Australian adolescents across sex [41,43]. The McDonald’s Omega of the scale was found to be 0.963 within the current study.

#### 2.2.4. Disordered Eating Behaviours

Binge eating and purging behaviours were measured with the eating disorder behavioural frequency questions from the Eating Disorder Examination Questionnaire (EDE-Q) [42]. This section of the EDE-Q asked participants to indicate how many times they have engaged in disordered eating behaviours over the last 28 days. Binge eating behaviours were measured with the following question ‘Over the past 28 days, on how many days have such episodes of overeating occurred (i.e., you have eaten an unusually large amount of food and have had a sense of loss of control at the time)’. Purging behaviours were measured by totalling the responses for questions examining the frequency of self-induced vomiting and laxative use as a measure of controlling weight and shape. As previously mentioned, the EDE-Q has demonstrated good reliability for Australian adolescent populations [43].

#### 2.2.5. Drive for Muscularity

The Drive for Muscularity Scale [44] consists of 15 6-point Likert-type scale items, allowing responses from 1 (never) to 6 (always), and asked participants to indicate their agreement with statements such as ‘I wish I were more muscular’. Scores are averaged and ranged from 1 to 6, with higher scores specifying a greater drive for muscularity. The scale has been shown to demonstrate good internal consistency and discriminant and convergent validity [44,45]. The McDonald’s Omega of this scale was found to be 0.921 within the present study.

#### 2.2.6. Psychological Distress

The Kessler Psychological Distress Scale (K10) [46] was used to measure the psychological distress of participants. This scale measured depression and anxiety symptoms over the past 4 weeks with 10 Likert-type scale items, from 1 (none of the time) to 5 (all of the time), and included questions such as ‘In the past 4 weeks, about how often did you feel hopeless?’. Items are added and total scores ranged from 10 to 50, with higher scores suggesting higher levels of experienced distress. Scores are classified as indicating an individual who is likely to be well (10–19) or experiencing a mild (20–24) or severe (30–50) mental disorder [47]. The K10 has good internal consistency and predictive validity for Australian adolescent samples [36,48] as well as in general samples [46]. The McDonald’s Omega of this scale was found to be 0.934 within the present study.

#### 2.2.7. Impairment

Three subscales from the Pediatric Quality of Life (PedsQL) Scale [49] were used to measure functional impairment. These included the physical functioning subscale, as well as the psychological functioning and social functioning subscales, which were combined into a ‘psychosocial’ subscale. These subscales consisted of 12 Likert-type items in total, from 0 (never) to 4 (almost always). The scale asked participants to indicate their agreement with statements such as ‘I feel scared or afraid’, ‘Other kids tease me’, and ‘I have low energy’. These scores are transformed onto a 0–100 scale, with higher scores indicating higher functioning. Although there are not commonly accepted clinical cutoffs for this scale, children suffering from cancer and cerebral palsy report a mean total score of 66–68, whereas healthy children report a mean total score of 84 [50]. The scale has demonstrated good internal consistency [36] and reliability and validity in adolescent samples [48,49,51]. The McDonald’s Omega of the physical and socioemotional subscales was found to be 0.852 and 0.902, respectively, within the present study.

### 2.3. Statistical Analysis

Analyses were conducted using IBM SPSS Statistics, version 24 (manufactured Armonk, NY, USA). The percentage of participants who reported that they had taken AAS within their lifetime was measured, along with a 95% confidence interval for the overall sample as well as within demographic sub-groups. All hypothesis testing was conducted with significance cutoffs of *p* = 0.05. Univariate analyses examining the demographic correlates of AAS use prevalence was conducted using binary logistic regression with biological sex, parents’ country of origin, sexual attraction, age, BMI percentile, body dissatisfaction, binge eating frequency, purging frequency, drive for muscularity, and weight and shape concerns as explanatory variables. Multivariate binary logistic regression was then used to investigate the relative strength and independence of the associations between each correlate and AAS use when adjusted for all other correlates. Assumptions were tested and were found to not be violated, including the assumptions of multicollinearity and logit linearity. Finally, one-way ANOVAs were used to compare K10 scores and PedsQL scores across adolescents who did and did not report lifetime AAS use. The assumption of homogeneity of variance, as measured through Levene’s test for equality of variances, was found to be violated across the physical and psychosocial subscales of the PedsQL (*p* < 0.001), as well as the total K10 scores (*p* < 0.001). Shapiro–Wilk analysis also revealed the distributions of the physical (*W* = 0.28, *p* < 0.001) and psychosocial subscales of the PedsQL (*W* = 0.882, *p* < 0.001) significantly departed from normality, alongside the distribution of K10 total scores (*W* = 0.886, *p* < 0.001). As such, Welch’s ANOVA was utilised for this analysis to account for these violated assumptions.

## 3. Results

### 3.1. Descriptive Statistics

In total, *N* = 5071 participants were included in analyses after excluding 207 responses for withdrawn or lack of provided consent (*n* = 92), high levels of missing data (*n* = 36), or non-serious responses (*n* = 79). As seen in Table 1, 46.1% reported their sex as male, 7.9% identified as Aboriginal or Torres Strait Islander, 28.2% stated that at least one of their parents were born overseas, and 12.6% reported same sex attraction or questioning sexual attractions. Age ranged from 11 to 19 years (*M* = 14.43, *SD* = 1.5), and BMI percentile ranged from 0.0 to the 99.5th percentile (*M* = 54.3, *SD* = 30.9). G*Power [52] was utilised to calculate that a conservative sample size of *n* = 4135 was required for a logistic regression analysis to calculate a 10% difference in probabilities of outcomes, with a power of 0.80 and significance level of 0.05. For a one-way ANOVA calculating a medium effect size of 0.25, a sample size of only *n* = 128 was required.

### 3.2. Prevalence and Distribution of Adolescent AAS Use

In total, *n* = 52 participants (1.1%; 95% CI = 0.8–1.4%) indicated that they had used AAS within their lifetime to increase their muscularity. Of these, *n* = 32 (62%) stated that they had used AAS without obtaining a prescription from a general practitioner/physician. Sensitivity analysis as seen within Appendix A revealed high demographic similarity between AAS users who reported having accessed prescribed vs. non-prescribed steroids. As such, the total AAS user group of *n* = 52 was utilised for maximal statistical power.

Table 1 displays the demographic distribution of lifetime AAS use. As can be seen, adolescents who reported AAS use to increase muscularity tended be male, same-sex attracted or questioning, and identified as Aboriginal or Torres Strait Islander.

### 3.3. Univariate Associations with AAS Use

Univariate logistic regressions examining factors associated with the likelihood of reporting lifetime muscularity-focused AAS use were conducted. As can be seen in Table 2, male adolescents exhibited 467% higher odds of using AAS in comparison to female adolescents, and adolescents who identified as Aboriginal or Torres Strait Islander showed a 280% increase in their odds of AAS use in comparison to non-Indigenous participants. Furthermore, participants who were same-sex attracted or questioning showed a 217% increase in AAS use odds in comparison to opposite-sex attracted adolescents. Greater drive for muscularity and weight and shape concerns, and more frequent purging and binge eating were associated with a higher likelihood of having reported lifetime AAS use. Drive for muscularity was particularly potent, with every one-point increase on the scale of 1–6 linked to a 191% increase in odds of AAS use. Furthermore, each additional episode of purging or binge eating over the past month was associated with 11% and 9% higher odds of AAS use.

Age, migrant status, and BMI percentile were not significantly associated with the odds of lifetime AAS use.

### 3.4. Multivariate Associations with AAS Use

The multivariate binary logistic regression combining and adjusting for the effects of each of the correlates was statistically significant, χ2 (*df* = 10, *N* = 3863) = 117.30, *p* < 0.001, and demonstrated an Akaike Information Criterion (AIC) of 275.22. This statistic is quite large, indicating a poor fit to the data, but may be expected given the large number of variables included within the model. As seen in Table 3, the only variables to remain significantly and independently associated with lifetime muscularity-focused AAS use were severity of drive for muscularity and frequency of purging behaviours. This is as correlations between other variables would likely have been high, reducing significance at the multivariate level.

### 3.5. Distress and Impairment by AAS Use Groups

See Table 1 for the means and standard deviations of the scores on measures of psychological distress and impairment among adolescents who did and did not report lifetime AAS use. The results of the one-way ANOVAs revealed that adolescents with lifetime AAS use were experiencing significantly higher levels of psychological distress (*F* (1, 50.52) = 23.51, *p* < 0.001, η^2^ = 0.01), physical impairment (*F* (1, 30.08) = 18.56, *p* < 0.001, η^2^ = 0.02), and psychosocial impairment (*F* (1, 30.13) = 13.10, *p* < 0.001, η^2^ = 0.01) in comparison to adolescents who did not report a history of AAS use. Based on the above high effect sizes, it is likely that differences in AAS use across biological sex and sexual attraction are meaningful, but not statistically significant due to issues with low power.

## 4. Discussion

This study aimed to provide an update to the prevalence and correlates of adolescent AAS use for the purposes of increasing muscularity. We found a prevalence rate that was lower (1.1% CI: 0.8% to 1.4%) than previously reported rates of around 2.3–2.4% in this population [25,26,27]. However, it is worth noting that these studies utilised larger samples (*N* = 10,314 to 21,905) than the present study. Groups most at risk emerged as consistent with the previous literature, including males, ethnic minority and Indigenous youth, sexual minority youth, and adolescents already reporting higher levels of concerns and behaviours associated with weight, shape, and muscularity. Support for considering AAS as a clinically significant problem behaviour was provided by our findings of association with higher levels of distress and functional impairment across physical and psychosocial domains.

The reduced prevalence rates for lifetime AAS use in comparison to other studies across Australian and global populations is worthy of consideration. While it is possible that rates are decreasing, this would be contrary to findings in other studies of increased muscularity dissatisfaction among young people and associated behaviours, there are alternative methodological explanations. For instance, adolescents in the EveryBODY study completed surveys at school among peers and were supervised by a school staff member. Given the illicit nature of anabolic steroids, social desirability bias and fear of reprimands from authority may have played an active role in under-reporting this behaviour. As such, actual AAS use within this sample may be higher than what was reported. It is important to consider the potential for ‘hidden users’ within future studies, especially within samples susceptible to social desirability bias, which could have implications for the assessment of use and the effective targeting of treatment. The counter argument to this is that the survey was anonymous and completed under exam-like conditions. Furthermore, as mentioned within previous school studies, conducting a cross-sectional design across only a school-based adolescent population excludes students who have not finished school and have opted to leave at a grade earlier than Year 12 [26]. It may be the case that these students have easier access to steroids or are more likely to use steroids as a result of being outside the school system. Perhaps if the current study had examined a population of adolescents outside of only a school-based setting, higher prevalence rates of AAS may have been reported. It is also true that access to AAS differs across countries, and thus the generalisability of these findings to other parts of the world, especially those less similar culturally to Australia, is uncertain.

Higher drive for muscularity and engaging in more frequent purging behaviours were identified as being associated with increased odds of adolescent AAS use above and beyond other variables (including biological sex, age, BMI, first- or second-generation migrant status, sexual orientation, Aboriginality, weight and shape concerns, and binge eating behaviours). The significance of drive for muscularity aligns with the literature, proposing that the pursuit of a muscular body ideal is a core driver for AAS use [24,37], which is particularly salient given the high proliferation of muscular body ideals within society [2] and the commonality of muscle building behaviours [3,9]. Furthermore, purging associations align with research demonstrating high rates of bulimia nervosa in competitive bodybuilders [53], as well as other abnormal eating behaviours [54]. Purging is an extreme weight control behaviour and is associated with greater eating disorder severity and worse prognosis [55,56]. Considering the similar motivations between eating disorders and steroid use in reducing body dissatisfaction, if one is willing to use extreme methods like purging to achieve an ideal body, they may also be more willing to use physique altering substances to push their body to the extreme to achieve these goals.

The above is supported by significant associations between more binge eating behaviours and higher levels of weight and shape concerns with increased odds of AAS use. This demonstrates that, similar to eating disorders, AAS use may be motivated by preoccupation and dissatisfaction with one’s own body shape [29]. Body dissatisfaction has been shown to result in distress and lead to behaviours motivated towards altering one’s body shape to more closely approximate societal ideals [5,9]. Indeed, research in other populations has also evidenced AAS use to be associated with increased risks of endorsing eating disorder symptoms [29,32]. The clustering of AAS use with eating disorder symptoms lends credit to the suggestion that AAS use is an extension of disordered eating behaviours [39].

We found male sex to be associated with higher likelihood of AAS use, which is in keeping with previous studies in other populations [19]. Although the ideal female body image is becoming more muscular [1,35,57], this ideal image favours a toned but slim build [1]. As AAS use can result in masculinising side effects [58] and a bulky male mesomorph shape, this may deter females from engaging in AAS use. Despite this, as more females engage in sports at an elite level, especially those favouring muscular physiques (e.g., Rugby League and Australian Rules football), this may change over time and as such should be mapped accordingly.

Identifying as same-sex attracted or questioning was also found to be associated with increased odds of AAS use. This could be attributed to the minority stress model, which posits that minority communities face increased social stress due to victimisation and stigma, increasing risk of substance use and poor health outcomes [33,59]. This concept of minority stress may also be compounded through greater internal group pressure within the gay and bisexual community to obtain a muscular physique [60], thereby resulting in even greater body dissatisfaction, motivating AAS use [33]. Differences across sexual orientation may also be accounted for by objectification theory [61] which posits that Western women’s bodies are considered objects for men’s pleasure, which can lead to internalisation through self-objectification and considering one’s perceived attractiveness. This can then lead to body shame, dissatisfaction, and disordered eating behaviour [62]. As such, it has been proposed that in attempting to attract male partners, gay men also experience self-objectification and body image disturbance, similarly to heterosexual women [63,64], which contributes to higher rates of maladaptive eating behaviours [65]. This is despite specifically desired body shapes differing between the sexes. In summary, AAS use within this community may be a direct response to body dissatisfaction from self-objectification and efforts to attract male partners. Although power is not sufficient within this study to test for an interaction between sex and sexual attraction to validate this theory, this could be a useful avenue for future research.

Although it was hypothesised that the location of one’s parents’ birth being outside of Australia would be associated with increased AAS use, this was not found to be an association within the current sample. Several previous studies have established clear associations between Asian, African American, and Hispanic adolescents within North American samples and increased likelihood of using AAS and exhibiting muscle enhancing behaviours [7,29]. These associations have been explained as adolescents’ attempts to achieve status, recognition, and acceptance within the majority western culture which emphasises strong and muscularised bodily physiques that differ from their own conceptualisations of body image [7,29]. As such, it may be the case that this is also reflected within the current sample, but many different cultures were included together in the same variable, which may have diminished any potential individual cultural associations.

On the other hand, we did observe higher rates of AAS use among adolescents identifying as Aboriginal or Torres Strait Islander. Colonisation of Australia is relatively recent and has replaced traditional cultural relationships with food, movement and bodies, and “Westernised” beauty ideals that typically exclude Indigenous representation [66]. It is suggested that media has exerted a significant influence upon Indigenous peoples worldwide, including within Australia, leading to a higher prevalence of overvaluation of weight and shape and eating disorders [67], as well as a desire to be bigger and more ‘built up’ [68]. However, others have suggested that the strengthening of cultural identity within Indigenous groups serves as a protective factor in promoting body acceptance, by shifting away from Westernised body ideals and towards holistic Indigenous models of health and wellbeing [66] and promoting cultural pride [69]. As such, the fostering of cultural engagement within adolescent Indigenous groups may be an effective way of protecting against the influence of body dissatisfaction, which may serve as a driver for the use of AAS within this population. Social media has been proposed as a useful vehicle for promoting positive Indigenous views of health [70], especially to Indigenous youth [71], and as such could be the ideal method of disseminating Indigenous body pride. Despite the above findings, however, a paucity of research on muscularity-focused body ideals conducted in non-WEIRD samples has been identified [72,73], especially within Indigenous Australian populations. This gap must be addressed within future research, ideally through a decolonising approach, prioritising Indigenous voices and research methodologies where possible.

Adolescents who reported AAS use were also found to have increased distress, as well as more physical and socioemotional impairment in comparison to non-AAS users. This aligns with previous research that has demonstrated an association between probable muscle dysmorphia and AAS [36], the deleterious physical health side effects of AAS [22,23], and increased risks of mood, anxiety, and behavioural disorders characterised by aggression, anger, and irritability [15,74,75]. Other studies have also reflected AAS-associated impairment, including increased rates of occupational disruption [74] and increased executive dysfunction [76]. Taken together, these findings demonstrate AAS use is a clinically significant behaviour warranting appropriate public health and potentially clinical attention. Despite this, it is worth noting that causality cannot be inferred from cross-sectional designs such as those utilised within this study.

Several clinical implications are apparent from the above findings. With information about factors associated with adolescent AAS use, clinicians now have updated information on the prototypical AAS user. This should help clinicians identify AAS use in at-risk clients and uncover how this substance use may be contributing to an individual’s presentation. This is especially important given the reluctance of AAS users to report AAS use and its side effects to health practitioners [20,21]. Regarding prevention, given the link we observed with eating disorder psychopathology, information about AAS use and its negative side effects could be incorporated into existing body image and eating disorder interventions already utilised in schools. It is hoped that this would be effective in raising awareness of AAS-associated harms among adolescents before AAS use is initiated to counteract popular perceptions of the safety of AAS use [14] promoted in online forums and through social networks [20].

The present study had several strengths, including that it analysed the data obtained across diverse genders and ethnicities from a large scale, school-based population, representative study involving adolescents of various ages. Furthermore, questions examining steroid use were asked in such a way as to increase construct validity by ensuring that the use was for increasing muscularity to rule out use of steroids for pre-existing medical conditions, which has been shown to artificially inflate AAS usage rates in previous studies [77]. However, several weaknesses may also be identified. Firstly, the present data analysis was cross-sectional, and as such, causality or the directions of associations cannot be determined. Secondly, no qualitative information about individuals’ own motivations for steroid use was gathered; therefore, the phenomenological experience of Australian adolescent steroid use remains under-explored. Finally, the grouped school setting of the present study makes responses vulnerable to social desirability bias and participant cross-contamination, as surveys were completed within class time under the supervision of teachers and school staff. This may have negatively influenced the true reporting of participants’ experiences as children were reluctant to disclose AAS use for fear of punishment or judgement from peers (despite the fact the students were informed the survey was anonymous)—providing a reason for why a lower prevalence rate was observed. Future studies investigating this adolescent population may benefit from providing students with links to surveys to complete at home, or within an environment external to school, to control for this potential social desirability bias. Additionally, it is suggested that future studies employ longitudinal designs to investigate the direction of causality of the above-mentioned AAS use correlates. Furthermore, it is suggested that qualitative studies be conducted to investigate the subjective experience and motivations of adolescent AAS use and to establish whether these align with the correlates highlighted in this study. A replication of this study with more participants would allow for an analysis of the interaction between sex and sexual attraction, to test for the impact of self-objectification in same-sex attracted males in contributing to AAS use. Finally, future studies could investigate the developmental trajectory of AAS use amongst adolescents to determine better at which age groups AAS use is likely to be initiated and increase, which may assist in efforts to capture problematic use early. This would be especially illuminating considering the mixed evidence of age and AAS use, as although there is some evidence to suggest an increase in muscle building behaviours and AAS use in older adolescents [7], other studies have reported pre-teen AAS use [24] or higher usage rates in younger adolescents [26].

## 5. Conclusions

In conclusion, this study explored updated prevalence rates and correlates of adolescent AAS use in an Australian population. The findings suggest that several demographic and body image related factors, alongside disordered eating behaviours, are associated with increased odds of AAS use. As such, this study provides information to assist with the identification, prevention, and treatment of adolescent AAS use and serves as a foundation for future longitudinal and qualitative research to further elaborate on motivations for adolescent AAS use.

## Figures and Tables

**Table 1 nutrients-17-00980-t001:** Demographic distribution of androgenic–anabolic steroid (AAS) use.

		Lifetime AAS Use			
		No		Yes	
		*n*	*%*	*N*	*%*
Biological Sex					
Male		1993	97.9	43	2.1
Female		2373	99.6	9	0.4
Parents born overseas					
Yes		1322	98.8	16	1.2
No		3378	99.1	31	0.9
Aboriginal or Torres Strait Islander					
Yes		364	96.8	12	3.2
No		4372	99.1	38	0.9
Same sex attracted or questioning					
Yes		537	97.1	16	2.9
No		3813	99.1	36	0.9
	Range	M	SD	M	SD
Age	11–19	14.43	1.6	14.78	1.5
BMI percentile	0–99.5	54.2	30.8	62.9	29.4
Weight and Shape subscale	0–6	1.60	1.78	2.54	1.99
Binge Eating Behaviours	0–28	1.50	4.56	6.02	9.86
Purging Behaviours	0–56	0.63	3.34	10.92	18.53
Drive for Muscularity Scale	1–6	1.92	0.93	3.74	1.74
PedsQL Physical	0–100	86.20	18.27	56.33	38.47
PedsQL Psychosocial	0–100	76.91	22.79	48.40	36.99
K10	10–50	20.85	10.03	30.80	14.63

Note: BMI = body mass index; PedsQL = pediatric quality of life scale; K10 = Kessler psychological distress scale.

**Table 2 nutrients-17-00980-t002:** Univariate logistic regressions.

Variable (Reference)	Wald Chi Square	AIC	b	SE (b)	*p*	OR [95% CI]
Male sex (female)	30.162	13.63	1.74	0.37	<0.001 *	5.67 [2.77, 11.70]
Age	2.18	44.14	0.14	0.09	0.14	1.15 [0.96, 1.37]
BMI percentile	2.94	296.65	0.01	0.01	0.10	1.01 [1.0, 1.02]
Aboriginal or Torres Strait Islander (non-Aboriginal)	12.35	13.78	1.33	0.34	<0.001 *	3.80 [1.97, 7.32]
Migrant status †	0.78	13.88	0.28	0.31	0.37	1.32 [0.71, 2.42]
Same-sex attracted or questioning (opposite-sex attracted)	12.04	14.01	1.15	0.30	<0.001 *	3.17 [1.74, 5.73]
Weight and shape concerns	13.29	159.44	0.25	0.07	<0.001 *	1.28 [1.12, 1.46]
Drive for muscularity	113.72	149.26	1.07	0.10	<0.001 *	2.91 [2.41, 3.51]
Binge eating behaviours	24.88	70.20	0.09	0.01	<0.001 *	1.09 [1.06, 1.12]
Purging behaviours	71.34	90.93	0.10	0.10	<0.001 *	1.11 [1.08, 1.13]

Note: CI = confidence interval; OR = odds ratio; * = *p* < 0.05; † defined as at least one parent born outside of Australia.

**Table 3 nutrients-17-00980-t003:** Multivariate logistic regression.

Variable	B	SE (b)	*p*	AOR [95% CI]
Drive for muscularity	0.89	0.15	<0.001 *	2.44 [1.81, 3.28]
Purging behaviours	0.10	0.02	<0.001 *	1.10 [1.06, 1.15]
Biological sex	1.06	0.56	0.06	2.89 [0.96, 8.64]
Same-sex attracted or questioning	0.73	0.51	0.15	2.08 [0.76, 5.70]
Aboriginality	0.41	0.58	0.48	1.50 [0.48, 4.70]
Weight and shape concerns	0.09	0.12	0.48	1.09 [0.86, 1.40]
Age	0.09	0.14	0.51	1.09 [0.84, 1.43]
Parents born overseas	−0.11	0.48	0.82	0.90 [0.35, 2.30]
Binge eating behaviours	0.00	0.03	0.92	1.00 [0.93, 1.06]
BMI percentile	0.00	0.01	0.99	1.00 [0.99, 1.01]

Note: CI = confidence interval, AOR = adjusted odds ratio; * indicates significant effect with *p* < 0.05.

## Data Availability

The data presented in this study are available on request from the corresponding author due to privacy and ethical considerations (see the above paragraph re: waiver of consent).

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
