# Peer review of "Prevalence and Correlates of Anabolic–Androgenic Steroid Use in Australian Adolescents"

_nutrients, 2025, doi:10.3390/nu17060980_

Round 1

Reviewer 1 Report

Comments and Suggestions for Authors

I have reviewed the attached article and would like to thank the editor and the authors for the opportunity to review this work. The manuscript is well-structured and addresses a highly relevant topic in the field of public health and clinical psychology, specifically the use of anabolic-androgenic steroids (AAS) among Australian adolescents. The study provides updated data on prevalence and correlates, making a significant contribution to the existing literature.

The aspects that enhance this study include:

  • The large sample size (5071 adolescents), which strengthens the reliability of the findings.
  • The use of validated psychometric instruments, such as the Drive for Muscularity Scale and the Kessler Psychological Distress Scale.
  • The in-depth discussion on the psychological and demographic correlates of AAS use, offering valuable insights for prevention and early intervention.
  • The emphasis on the need for further qualitative and longitudinal studies to better understand the phenomenon of AAS use among adolescents.

However, some sections require improvements:

  • Lines 301-310: The explanation regarding the lower prevalence rate compared to previous studies could be further elaborated, taking into account possible variations in cultural context or access to AAS.
  • Lines 373-375: The lack of a significant association between parental birthplace and AAS use should be better justified by analyzing specific cultural factors that might influence adolescent behavior.
  • Lines 399-405: The association between AAS use and high levels of psychological distress and functional impairment is robust, but a more in-depth comparison with previous studies would help contextualize the findings.
  • Lines 427-431: The impact of social desirability bias on underreporting AAS use could be discussed in greater detail, highlighting strategies to mitigate its effects in future studies.
  • Lines 435-442: It would be useful to specify at which age groups AAS use tends to increase to better target prevention strategies.

Regarding the study’s limitations, if they are not explicitly mentioned, the following should be considered:

  • The cross-sectional design, which prevents establishing causal relationships between the investigated factors and AAS use.
  • The potential social desirability bias, which may have led to underreporting steroid use.
  • The lack of qualitative data, which could provide a deeper understanding of the personal motivations behind AAS use.

The following articles should be cited to enrich the discussion and strengthen the study’s context:

  • Diotaiuti et al. (2022): Psychometric properties and measurement invariance across gender of the Italian version of the tempest self-regulation questionnaire for eating adapted for young adults. This article could be included in the methodology or study limitations section to highlight the importance of using validated measurement tools to assess eating behaviors and weight regulation.

It is suggested that these citations be inserted in the discussion section when analyzing the implications of the findings for prevention and early intervention. I would like to express my gratitude once again for the opportunity to review this study, which represents a valuable contribution to understanding AAS use among adolescents.

Full Citations:

  • Diotaiuti, P., Girelli, L., Mancone, S., Valente, G., Bellizzi, F., Misiti, F., & Cavicchiolo, E. (2022). Psychometric properties and measurement invariance across gender of the Italian version of the tempest self-regulation questionnaire for eating adapted for young adults. Frontiers in Psychology, 13, 941784. https://doi.org/10.3389/fpsyg.2022.941784
Comments on the Quality of English Language

The quality of the English language in the manuscript is generally strong, with clear and coherent expression of ideas. The text is well-structured, with appropriate academic tone and terminology. However, there are some areas where improvements can be made to enhance clarity and readability:

  1. Grammar and Syntax: Some sentences are overly complex and could benefit from simplification for better readability. For example, in lines 301-310, the explanation about the prevalence rate could be more concise to avoid excessive subordination.

  2. Word Choice and Precision: In certain sections, word choice could be more precise to avoid ambiguity. For instance, lines 373-375 discuss the absence of association between parental birthplace and AAS use, but the explanation could be more explicit in distinguishing between cultural and statistical factors.

  3. Consistency in Terminology: The manuscript uses various terms to describe similar concepts (e.g., drive for muscularity vs. muscle dissatisfaction). Ensuring consistency in terminology throughout the paper would improve cohesion.

  4. Use of Passive Voice: While academic writing often employs passive voice, excessive use can make sentences less direct. Consider revising sections where the subject of the action is unclear, such as in lines 399-405, to enhance readability.

  5. Typographical and Minor Errors: There are occasional minor typographical errors (e.g., line 472, "stidy" instead of "study"), which should be corrected through careful proofreading.

Reviewer 2 Report

Comments and Suggestions for Authors

I've got the opportunity to read and review the study about the anabolic in the Australian cohort. I found this study interesting with several implications for sports practitioners. However, some areas must be improved. Please find detailed comments below.

Major comments:

  • this is an unsupervised cross-sectional study. Therefore, double check whether you applied all the STROBE criteria. Especially see that you omitted the exact study recruitment period.
  • I strongly suggest to provide more information about the schools from which you extracted data. Perhaps, another table in the supplementary file would be welcome.
  • Reconsider to describe inclusion and exclusion criteria. Currently, you have to transfer to another study which could not be available for all (eg. Due to the open-access requirements).
  • Attach this survey in a supplementary file.
  • The description of statistical analysis needs more transparency. Provide manufacture location for your software and p-value borderline. Moreover, did you tested the basic data normality and how you present your descriptive statistics?
  • You recruited some participants, but I’m missing any information about your sample power. For example G*Power could help, but if you applied different methods, please describe.

Minor comments:

  • Revise the background of your abstract. State clearly the aim and avoid providing the period since there were no studies in this area.
  • I’m missing the exact results in your abstract. Provide statistics to support description.
  • Do you have any purpose to highlight the time period in conclusions or in the abstract? See where you wrote „since 2011” and „the first in 19 years”?
  • State your conclusions more clearly and precisely. Describe which factors and how are associated with AAS use (increase/decrease/aggravation/alleviation etc.).

To sum up this report, I recommend revisions. The authors should especially focus on major comments.

Round 2

Reviewer 2 Report

Comments and Suggestions for Authors

The authors applied necessary revisions.